# Household Drug Management Practices of Residents in a Second-Tier City in China: Opportunities for Reducing Drug Waste and Environmental Pollution

**DOI:** 10.3390/ijerph18168544

**Published:** 2021-08-12

**Authors:** Yumei Luo, Kai Reimers, Lei Yang, Jinping Lin

**Affiliations:** 1School of Business and Tourism Management, Yunnan University, Kunming 650106, China; luoyumei@ynu.edu.cn (Y.L.); 17713576252@163.com (L.Y.); 2School of Business and Economics, RWTH Aachen University, 52062 Aachen, Germany; reimers@wi.rwth-aachen.de; 3School of Resource Environment and Earth Science, Yunnan University, Kunming 650106, China

**Keywords:** household drug management, practice, environmental health, Chinese residents

## Abstract

The total amount of drug waste is expanding significantly as populations age and societies become wealthier. Drug waste is becoming a problem for health and the environment. Thus, how to reduce and effectively dispose of drug waste is increasingly becoming an issue for society. This study focuses on household drug management, which involves five sub-practices: selection, purchasing, using, storing, and disposing of drugs. A questionnaire survey was conducted in a second-tier Chinese city that reveals both problems and opportunities in these five sub-practices. The results show that consumers are aware of significant issues with regard to the safe and effective use of drugs as well as with regard to proper ways of disposing of and recycling drugs. Moreover, our analysis reveals promising opportunities for addressing these issues by developing novel services based on the idea of connecting the five involved sub-practices of household drug management. Connecting and adjusting practices in this manner can be seen as an important factor in reducing drug waste and pharmaceutical pollutants.

## 1. Introduction

With the continuous rapid development of the global pharmaceutical industry, global pharmaceutical spending is expected to grow by 2–5% per year to more than $1.1 trillion by 2024, with annual production reaching more than 100,000 tons [1]. However, at the same time, waste pharmaceuticals in households are also increasing. For example, two-thirds of the drugs sold in the United States are unused, costing the country between 2.4 and 5.4 billion USD [2]. In Australia, the estimated value of waste drugs per patient is about AUD 1308 per year [3]. In China, about 78.6% of families have the habit of storing drugs and more than 80% of families do not regularly purge their stock of expired drugs and even fewer people know how to properly dispose of the expired medicine [4]. 15,000 tons of expired drugs are produced annually [5]. In fact, medication waste residing within homes has negative effects on the economy, the environment, and even human health, especially when inadequate methods of disposal are used [6].

The existence of active pharmaceutical ingredients (API) in the environment is a global problem. A great diversity of commonly used pharmaceuticals, such as antibiotics, painkillers, hormones, and anti-inflammatory drugs, have been frequently detected in the environment, including water, sediments, soil, etc. in Canada, China, France, Sweden, and the United States [7,8]. Ongoing research produces more and more evidence that pharmaceutical ingredients in the environment can have adverse effects. Studies in the Potomac River Basin in the United States have shown that chemical drugs are inextricably linked to fish mortality and a high prevalence of intersex or testicular oocytes [9]. The psychotropic oxazepam and 17α-ethynylestradiol can, respectively, affect the sexual characteristics and habitat behavior of male fish [10,11]. In addition, antibiotic substances do not only have direct toxicological effects, but also facilitate the development and spread of antibiotic-resistant microorganisms, a severe threat for human health in the long run [1].

In the past decade, researchers in different fields, including environics, management science, and pharmaceutics, have made good progress in the minimization of the environmental load and ecological risks of pharmaceutical residues. Currently, decreasing the emissions from sources of pharmaceutical pollution from a drug administration perspective is considered a promising area of research [8,12]. Pharmaceutical management, according to the WHO, aims to reduce financial expenditure, avoid waste, increase access, and ensure that drugs are properly used, and involves multiple phases, including drug selection, procurement, storage, distribution, and use [13]. In the household context, drug management is also associated with disposal of unused or expired drugs. However, to date, the practices of drug management in the household context are still unclear.

The commonly recognized sources of pharmaceutical pollution include the production of household drug waste and the disposal of leftover drugs. Household drug waste is the result of a number of factors encompassing: (1) patients recovering before their dispensed medicines have all been taken; (2) therapies being stopped or changed because of ineffectiveness and/or unwanted side effects; (3) patients’ deaths; and (4) factors relating to repeat prescribing and dispensing processes [14]. The importance of public practices regarding disposal of unwanted medicines as sources of entrance of pharmaceuticals into the environment has been pointed out [15]. Thus, effective household drug management practices concern not only reducing drug waste to improve the use of drugs, but also the prudent disposal of expired/unused drugs [16]. In sum, both reducing preventable medicine waste and prudently disposing of inevitable medicine waste should be an obvious starting point of reducing the introduction of pharmaceutical residues into the environment.

Many studies have focused on drug disposal practices [15,17,18,19,20,21]. However, few studies have identified or accounted for drugs currently stored and used in households [22]. Moreover, no study has yet looked at the combined constellation of drug disposal and other drug management practices, such as selection, purchase, storage, and use. Therefore, the present study considers household drug management practices in terms of five sub-practices: selection, purchasing, use, storage, and disposal and was carried out among Chinese households to assess them. The findings provide detailed insights into household drug management practices and present a number of problems and opportunities with regard to both better waste drug management and safer and more effective medication management. The findings of this study provide evidence to inform health policy and medication practice with regard to reduction of household drug waste and environmental pollution.

## 2. Household Drug Management

Household drug management (HDM) is performed by consumers and includes selection, purchasing, use, storage, and disposal of drugs (see Figure 1). These five sub-practices in HDM are interlinked and interact with each other.

### 2.1. Selection

Selection of drugs is about ‘how’. In a household drug context, the selection sub-practice refers to how consumers choose drugs to treat their illnesses. The selection of the drugs is based on the illnesses. When a consumer feels ill, he/she will first make a preliminary judgment about his/her illness and then decide whether to see a doctor or to self-medicate. In the former case, drugs are selected according to the doctor’s advice or prescription. However, in the latter case, what drugs are chosen mainly depends on the consumers’ experience and/or recommendations by family, friends, and sales personnel [23].

### 2.2. Purchasing

For consumers, the purchasing of drugs is based on decisions on which drugs to use and is strongly affected by economic factors. The purchasing of drugs concerns ‘which’. One concern is the selection of drugstores. In China’s first- and second-tier cities, there are many drugstores to choose from. Therefore, when a consumer decides to purchase one particular drug, he or she must choose from which drugstore to purchase it. A second concern is the selection of a manufacturer. There are often many different manufacturers for the same kind of drug. Consumers therefore also often have to make a choice regarding the manufacturer. Both these choices are made mainly with regard to the availability and price of drugs.

### 2.3. Use

Use of drugs is related to ‘how’. After having selected and purchased a drug, or alternatively after having selected a drug from the household inventory, consumers need to decide how to take the drug, how often and in what dosage to take the drug, whether and how to combine different drugs, and how to adjust the medication after a while. Decisions on drug use are directly related to the therapeutic effect of drugs on consumers [13].

### 2.4. Storage

From a consumer perspective, storage concerns ‘why’ and ‘how’. ‘Why’ refers to why consumers store medicines in their homes. In China, storing drugs for emergent use in homes in large quantities and varieties has become a common practice. Drugstores’ promotions, large package sizes of drugs, and overly generous prescriptions are further reasons for storing drugs in homes [24]. The ‘how’ concerns the question of how to store drugs properly in the home. Generally, household drugs should be kept in a fixed position for ease of management and the storage environment should be clean, cool, and dry. In order to prevent harm to consumers caused by the expiration or invalidation of drugs in the home, consumers should purge their stock from expired or invalid medicines frequently. Thus, it is an important activity to sort through and clear up the household medicine inventory regularly.

### 2.5. Disposal

For expired or unwanted drugs in their homes, consumers need to decide how to dispose of them. Proper disposal is of great importance for consumers’ health and the environment [22]. On the one hand, if consumers accidentally ingest expired or invalid drugs, it may worsen their illness or delay recovery. On the other hand, some drugs, such as antibiotics, blood thinners, and chemotherapy drugs, can become toxic and hazardous to human health and the environment if they are not disposed of properly [25]. It is urgent for consumers to know how to effectively dispose of these drugs. Thus, the disposal of drugs is about ‘how’.

### 2.6. The Interplay between These Sub-Practices

All these sub-practices are interconnected to form a whole household drug management practice. Hence, effective household drug management should consider the interplay between these sub-practices. For example, what drugs consumers buy may be influenced by the drugs stored in the home. Consumers may choose to buy complementary drugs instead of all drugs needed for treatment, which requires a connection between the selection and storage practices. Moreover, the connection between the use and the storage practices would be conducive to reducing waste. For example, if consumers would use drugs with nearer expiration dates first, the amount of expired drugs could be reduced significantly.

## 3. Materials and Methods

To understand the current situation of household drug management in China, a questionnaire survey was conducted over a two-week period in May 2019. The study was approved by the Committee on Human Subject Research and Ethics of Yunnan University (CHSRE number: CHSRE2021017).

### 3.1. Questionnaire Design

The questionnaire was divided into three sections. The first section concerns respondents’ demographics. The second section covers the factual situation concerning the selection, storage, and disposal of drugs in households. The third section seeks to identify certain problematic situations that consumers may face in their purchasing and use of drugs.

The initial draft of the questionnaire was developed based on published studies on use, storage, and disposal practices [19,26,27] and feedback from experts during a workshop about waste drugs recycling, medication management, and water held in Kunming on 21 September 2018. Participants included three managers, three professors, a clinician, a pharmacist, and two students. They were asked to comment on content clarity, relevance, validity, and conciseness of the items in the questionnaire. Before sending out the questionnaire, a draft of the questionnaire was trialed on a clinician, a senior professor, and two undergraduate students. Based on their feedback, minor modifications to the structure and wording of the questionnaire were made.

Two kinds of questions were used, those which concern the factual situation in households and those which describe certain situations and problems consumers may face. For the first kind, i.e., for the descriptive questions regarding the factual situation in households, respondents had to choose from multiple given answers; with regard to the second kind of questions concerning possible problematic situations, respondents had to indicate their degree of agreement with given statements on a five-point Likert scale which ranged from ‘strongly disagree (1)’ to ‘strongly agree (5)’ (see Table A1).

### 3.2. Data Collection

The data were gathered in a second-tier city in the Southwest of China (Kunming) in 2019. The size of the population of this city is about 6.85 million. The city is divided into five administrative districts. Respondents were evenly distributed across these five districts and the most populous residential areas of each district were selected to ensure that a high number of potential respondents were available. People on the streets in these districts were approached randomly and invited to participate. People were approached at different times during the day, on weekdays as well as during the weekends. Participants could either complete the questionnaire themselves or have their responses entered by the researchers (mainly older people with poor vision) and were paid CNY (Chinese Yuan) 15–20 as compensation for answering the questionnaire. Participants had to be over 18 years of age. Potential respondents were informed about the purpose of the study and assured that any data collected would remain confidential. 628 questionnaires were delivered. Invalid questionnaires with more than half the questions unanswered were excluded. 558 valid questionnaires were available for data analysis.

### 3.3. Data Analysis

The data collected were entered into a spreadsheet and then exported to SPSS20.0 (SPSS Inc., Chicago, IL, USA) for analysis. Descriptive questions were summarized by frequency counts, while situational questions were summarized through mean values. Based on a one-sample *t*-test, values significantly smaller than 3 (the hypothesized theoretical mean) were considered as indicating low agreement and vice versa. A *p*-value of 0.05 was considered to be statistically significant.

## 4. Results

### 4.1. Demographics of Respondents

Table 1 shows the demographic characteristics of the respondents. We compared the demographic information between the 558 valid participants and the 70 invalid ones, and identified no statistical differences in terms of age, education, or gender across the two groups.

### 4.2. The Current Situation of HDM in China

#### 4.2.1. Selection

When consumers decide to self-medicate, there are several ways to choose drugs, such as reliance on their own experience, recommendations by family members or friends, reliance on advertisements/Internet websites/newspapers and magazines, and recommendations by sales personnel. Figure 2 shows that more than half of the respondents rely on their own experience or recommendations by sales personnel when self-medicating.

#### 4.2.2. Purchasing

The purchasing sub-practice is mainly related to the selection of drugstores and drug manufacturers. The selection of drugstores mainly depends on whether a drugstore has the drugs consumers want in stock and the price of the drugs. It is common that similar drugs supplied by different manufactures are jointly displayed in drugstores for consumers to choose. Therefore, respondents were asked to relate to the following three situations, namely: P1 ‘I often wonder which drugstore has the medicine I want to buy’; P2 ’It is not always known which drugstore is cheaper, more affordable, or has promotional activities’; P3 ‘When faced with the same drugs produced by different manufacturers, I often do not know which one to buy ‘. One-sample *t*-tests show that the mean values for P2 and P3 are both significantly higher than our threshold value of 3 (MP2 = 3.22, *p* = 0.00; MP3 = 3.36, *p* = 0.00) (see Table 2). Therefore, P2 and P3 can be considered to indicate urgent problems within the purchasing practice. By contrast, P1 does not seem to indicate a serious problem for consumers.

#### 4.2.3. Use

Effective use of drugs includes how to take the drug, how often and in what dosage to take the drug, whether and how to combine different drugs, and how to adjust the medication after a certain point, especially with regard to long-term treatments. Respondents were asked about three possibly problematic situations concerning their use practice, namely: U1 ‘How to take the drug’; U2 ‘How to adjust the medication’; and U3 ‘When to take the drug’. Table 3 shows that the mean value for U1 was significantly less than 3 (MU1 = 2.68, *p* = 0.00), for U2 significantly higher than 3 (MU2 = 3.15, *p* = 0.00), while the value was non-significant for U3 (MU3 = 3.06, *p* = 0.17), indicating that U2 is considered problematic.

#### 4.2.4. Storage

Regarding the storage practice, the first thing to know is what causes consumers to store drugs in their homes and how many drugs are stored in general. As shown in Figure 3, 72.8% of respondents said that they have a habit of storing drugs in their homes. Figure 4 shows that 18% of respondents are unclear about this number, while 27% store more than 20 boxes or bottles in their homes. This suggests that storing large amounts of drugs is a common practice in China.

Second, how and where to store drugs is also an important question. Figure 5 shows that, although 50% of respondents chose a fixed place, only 41% of respondents actually store drugs in a clean, cool, and dry place while 9% of respondents store drugs in various places in their homes. Therefore, it can be said that the storage practice currently poses an additional risk for the safety of consumers and possibly for the environment too because drugs that are improperly stored may also become unusable and thus need to be discarded prematurely.

Third, regular and timely purging of household drug cabinets is helpful for consumers to know a drug’s status and to effectively avoid expiration or invalidation of drugs. Figure 6 depicts the current situation. Only 25% of consumers regularly purge their drug cabinets while most consumers (64%) do so irregularly; 11% have never cleared up their drug cabinets. Figure 7 shows that 50% of consumers reported having expired drugs in their homes while 19% said that they did not know whether they had expired drugs in their homes.

#### 4.2.5. Disposal

How can we dispose of expired or unused drugs? In principle, there are two methods: recycling and proper disposal. 67% of respondents have never heard of expired drugs recycling activities, 29% have only heard about such activities, and very few (4%) have actually taken part in recycling activities, as shown in Figure 8. Regarding the disposal of expired drugs, 83.2% of respondents throw expired or unused drugs into the household garbage while 8.2% do send them to recycling points (see Figure 9). The study also asked respondents to indicate the extent to which they agree with this problematic situation: ‘When preparing disposal of expired drugs, I do not know how to do it effectively’. The associated mean value of 3.14 and the result of one-sample *t*-test (*t* = 2.958, *p* = 0.003) suggest that consumers generally feel a need for better information regarding proper methods of drug disposal.

## 5. Discussion

Our analysis has exposed significant problems with regard to the question of how to accurately select and purchase necessary drugs and how to safely and effectively store and use drugs in households, as well as how to effectively and properly dispose of and recycle expired and unused drugs.

Our survey has identified a number of problems and opportunities with regard to both better waste drug management and safer and more effective medication management (see Table 4). In particular, consumers tend to buy drugs in excess of their immediate needs in order to ‘replenish’ their family drug cabinet. In addition, large drug package sizes and direct-to-consumer (DTC) promotions contribute to large family drug inventories. These inventories are used mostly for self-medication which, in turn, creates the problem of how to select drugs. Here, consumers mostly rely on their own experiences or on the advice given by family members, friends, and sales personnel rather than by pharmacists and physicians, which can lead to the problem of improper treatment, such as improper drug use, incorrect dosage, antibiotic resistance, and adverse drug reactions, as well as the phenomenon of hoarding drugs [28]. In addition, sales personnel may persuade consumers to buy higher quantities of drugs in order to improve their own sales performance, which will also result in drug waste.

Moreover, even in the case of prescription drugs consumers feel at a loss when it comes to adjusting the medication if drugs are taken for a longer period of time. Due to adverse drug effects, the perceived absence of beneficial effects, inconvenience in dosing schedules, change in therapy as prescribed by their physician, or even a poor perception of the severity of their illness, patients will imprudently discontinue medications or self-adjust medication [29]. Poor adherence easily produces a large number of unused drugs [2]. Thus, effective medication management is essential to guide consumer drug selection and drug purchasing habits.

Previous studies have shown that the extent of unused medication in households is 42% in the United States [2], 31% in Sweden [30], and 72% in Portugal [26]. Additionally, 88% of respondents in Ireland have unused medicines at home [31] and 60% of respondents in Australia reported having unwanted medicines at home [17]. A total of 50% of respondents in our study declared that they keep expired drugs in their household. Thus, the presence of expired or unused drugs in homes is a widespread phenomenon throughout the world. Moreover, the number of drugs stored at home is related to adverse drug events (ADEs). Patients with a large inventory of drugs are more likely to experience therapeutic duplication [32].

More than half (59%) of respondents in the present study appear to practice inappropriate storage behavior, while only 25% regularly clear their medicine cabinets. In fact, there are strict requirements for the storage of drugs, and irregular storage can lead to the weakening of efficacy, oxidation, and even toxicity. Sorensen, Bpharm, Purdie, Fracp and Mba [32] found that patients who stored drugs in multiple locations were 4.2 times more likely to be associated with poor health outcomes. Keeping unnecessary or expired drugs at home is both a cause and a consequence of the irrational use of medicine [33].

Effective storage in households is related to other sub-practices, such as selecting and purchasing necessary drugs, and taking the appropriate dosage can reduce the quantity of drugs stored. Hence, reminding and informing patients about the storage conditions of drugs and the proper dosage can also reduce waste drugs.

Regarding disposal of waste drugs, consumers seem to be acutely aware of the problems expired or unused drugs can create for the environment and for society. However, the majority of respondents exhibited inadequate awareness and poor disposal practices. Three-quarters of our respondents disposed of drugs by discarding them together with the household waste, throwing them in the garbage, sink, toilet, etc., and only a small sub-group (8.2%) return expired drugs. Although environmental awareness may impact the choice of what is considered the safest means of disposal, actual behavior does not always live up to this awareness [34]. Other countries face similar problems [17,18,35]. Many studies have pointed out that inappropriate disposal constitutes an additional threat for human health and ecosystems [1,33,36].

Drug take-back programs aim to provide a safe and effective method of drug disposal by recycling unused and expired drugs and prevent excessive entry of APIs into the environment. Under the provisions of current European Union legislation, all EU member states must establish collection schemes to recover and safely dispose of unused and expired medicines [37]. In the United States, the Food and Drug Administration (FDA) offers consumer information about safe practices for drug disposal, and the Drug Enforcement Administration (DEA) regularly sponsors national activities to collect unused drugs [18]. In China, local governments and some enterprises have launched drug recycling activities. However, as in many countries, there are poorly functioning waste management schemes, insufficient funds and publicity, no legislation, and no obligation for pharmacies to participate in or promote these schemes [34]. Thus, a strict legal framework and well organized, cost-effective, and easily accessible state-run disposal systems are necessary in order to enable the general public to reduce the negative pharmaceutical impacts on the environment by returning unused pharmaceuticals to collection schemes for proper disposal.

Overall, our findings lend support to the idea that significant improvements can be realized if the five sub-practices are more tightly connected. On the one hand, there is a readiness and awareness among consumers to act on problems related to both waste drug management and medication management. On the other hand, consumers face significant problems with regard to both these issues. For example, consumers would benefit from professional advice on how to adjust long-term medication regimens to their ongoing condition. They would also appreciate professional advice on how to correctly dispose of and/or contribute to recycling of expired and unused drugs. Finally, they would also welcome better drug management that helps to avoid untimely or unnoticed expiration of drugs or, in cases where expiration cannot be prevented, gives replacements for expired drugs. Physicians and pharmacists, in turn, may continue to support their patients with regard to adjusting medication regimens and advising consumers on the proper use and disposal of drugs. Conversely, manufacturers and pharmacies may see a business opportunity in supporting such activities, e.g., by helping families to replace expired drugs at little or no cost. After all, consumers are concerned about the costs of medication as well as about having a well-stocked family pharmacy inventory.

The limitations of this study include the random “hit and miss” approach of selecting respondents, which might cause some biases. In addition, the study did not consider OTC and prescription drugs separately. Since the purchase and quantity of prescription drugs are limited by prescription, they may have different characteristics from OTC drugs in the practices of household drug management. Therefore, further comparative studies of household drug management practices for different drugs are needed.

## 6. Conclusions

This study assessed the practices regarding household drug management, namely selection, purchasing, use, storage, and disposal among 558 Chinese citizens and suggests that there are significant problems with regard to the question of how to accurately select and purchase necessary drugs, safely and effectively use drugs in households, and how to effectively and properly dispose of and recycle expired and unused drugs. Moreover, our analysis has also revealed promising opportunities for addressing these problems by connecting and aligning the five sub-practices of household drug management.

## Figures and Tables

**Figure 1 ijerph-18-08544-f001:**
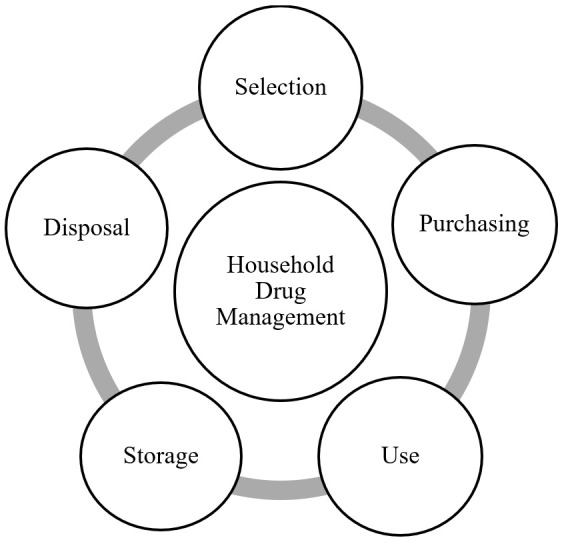
Household Drug Management.

**Figure 2 ijerph-18-08544-f002:**
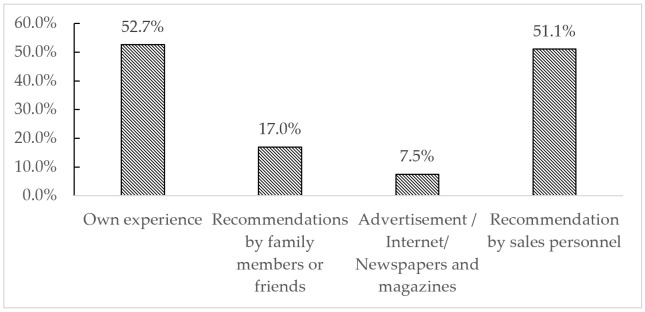
Ways of selecting drugs.

**Figure 3 ijerph-18-08544-f003:**
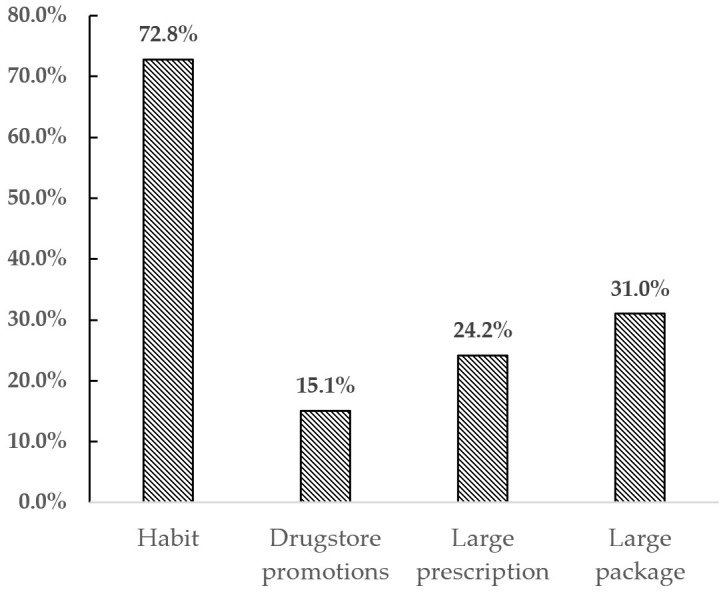
Reasons to store drugs in homes.

**Figure 4 ijerph-18-08544-f004:**
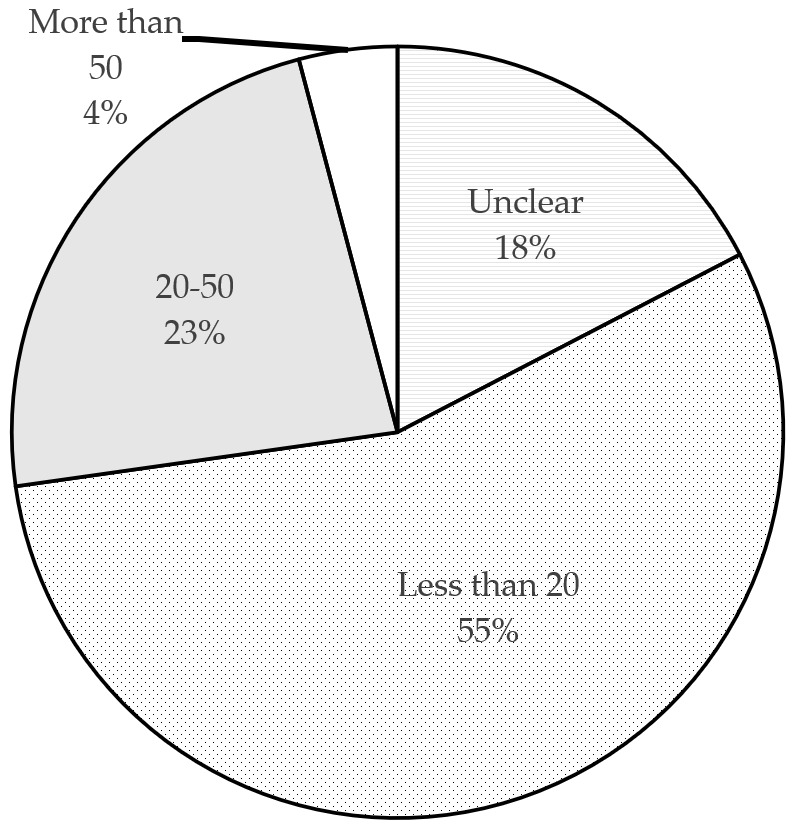
The number of drug packages stored in homes.

**Figure 5 ijerph-18-08544-f005:**
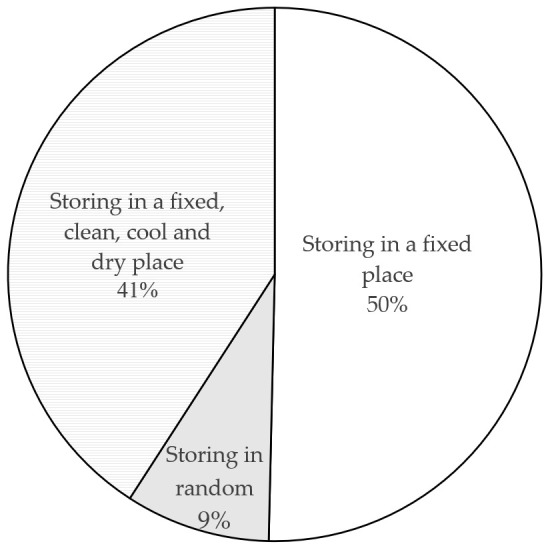
The situation of drug storage in homes.

**Figure 6 ijerph-18-08544-f006:**
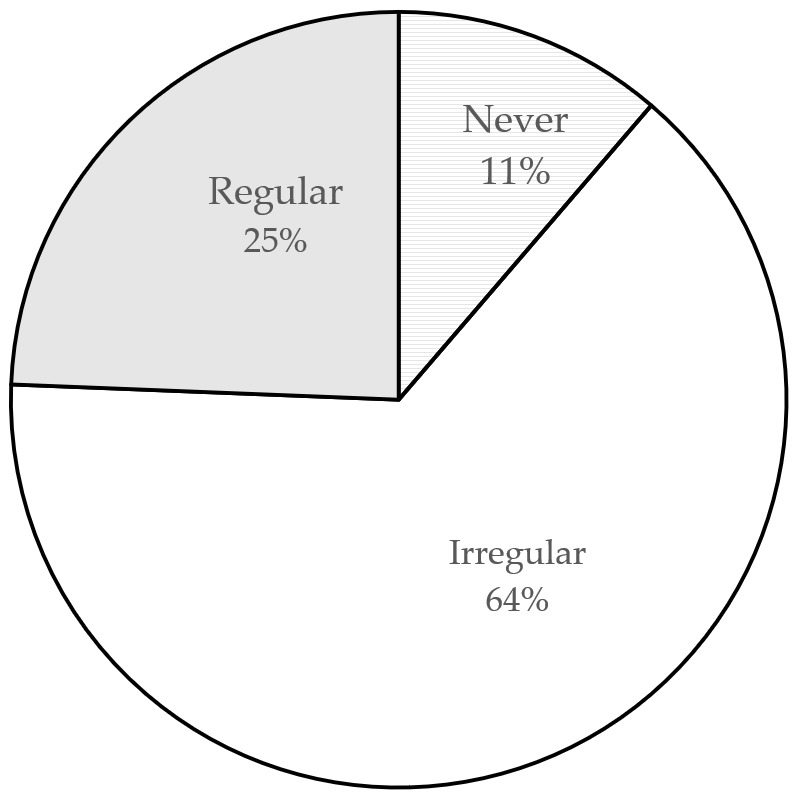
How often consumers clear up their drug cabinets.

**Figure 7 ijerph-18-08544-f007:**
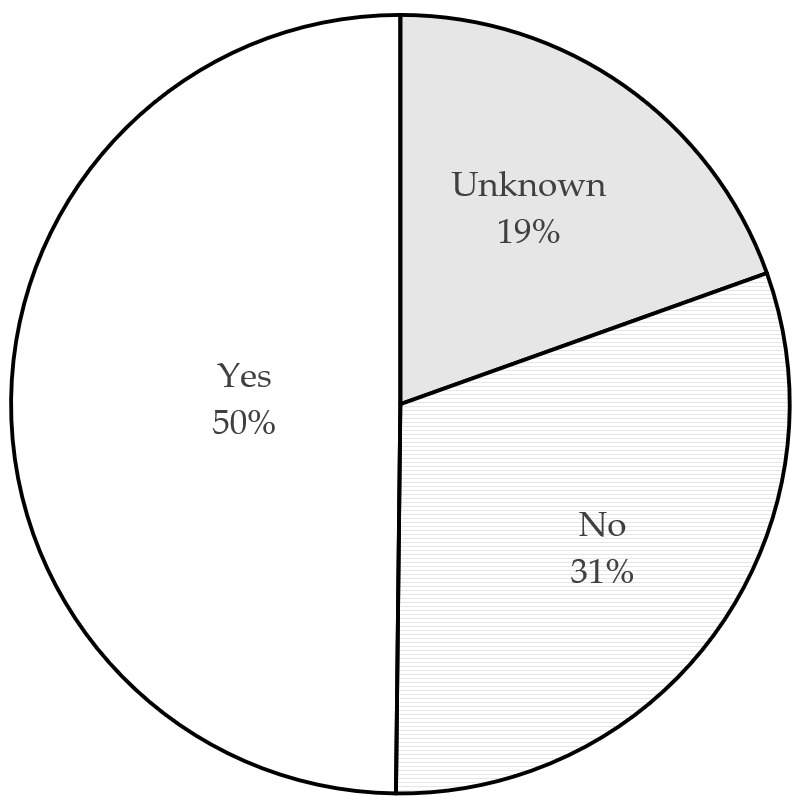
Percentage of respondents who store expired drugs in their homes.

**Figure 8 ijerph-18-08544-f008:**
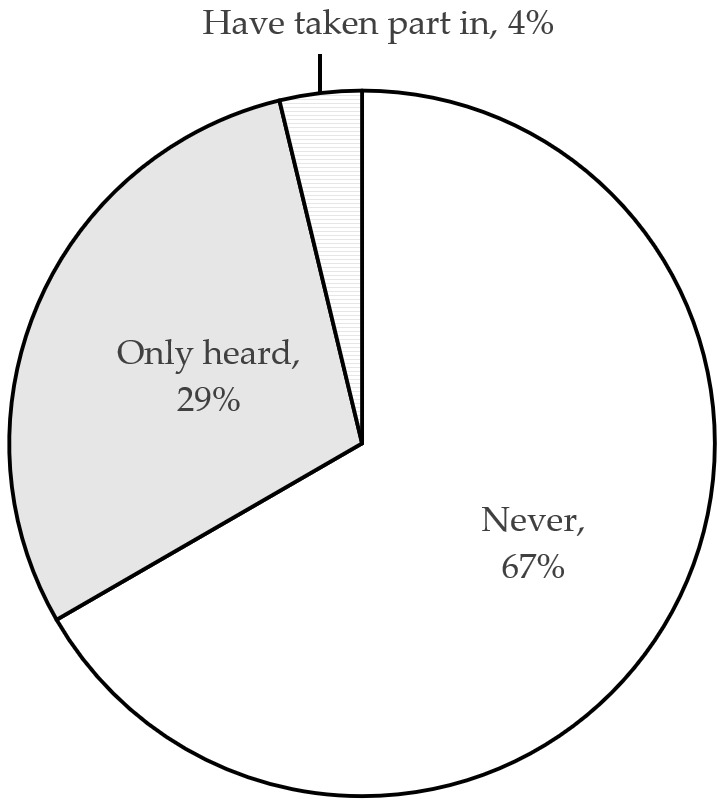
Awareness about the possibility of used drugs recycling.

**Figure 9 ijerph-18-08544-f009:**
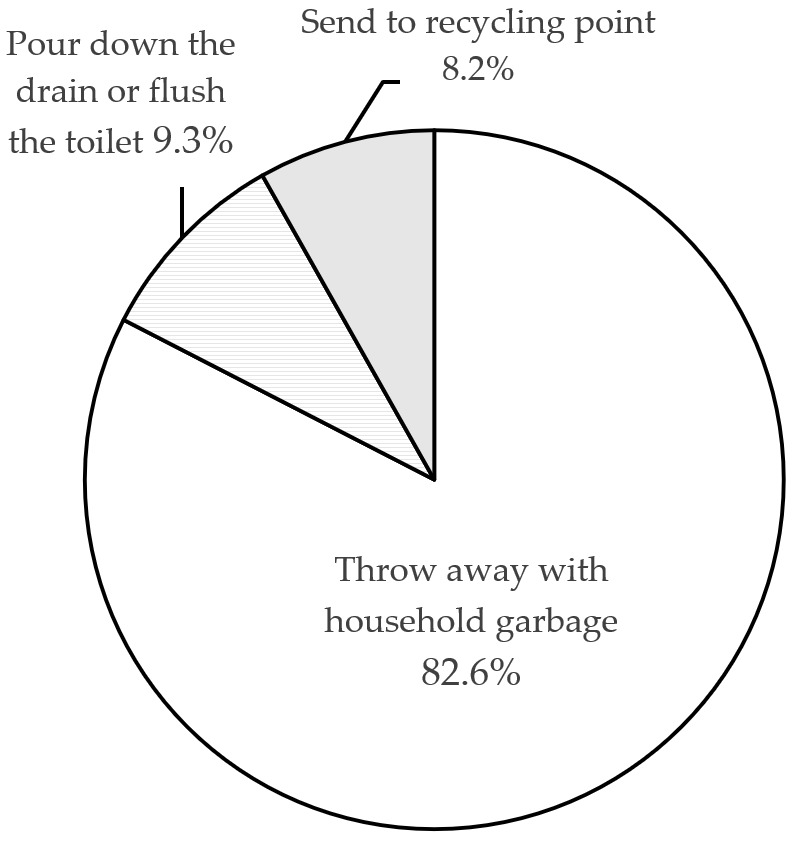
Methods of expired drug disposal.

**Table 1 ijerph-18-08544-t001:** Sample demographics (*n* = 558).

Measure	Items	Number of Respondents	Percentage %
Age (Years)	<20	52	9
	20–35	261	47
	36–50	172	31
	>50	73	13
Education	High school or below	168	30
	College or Bachelor	294	53
	Master or PhD	96	17
Income/per month (Chinese Yuan)	<2500	144	26
	2500–5000	207	37
	5000–10,000	147	26
	>10,000	60	11
Gender	Male	266	48
	Female	292	52
Number of family members	1–2	40	7
	3	177	32
	4	172	31
	5	169	30
Administrative districts	Wuhua	122	22
	Chenggong	102	18
	Panlong	103	19
	Guandu	117	21
	Xishan	114	20

**Table 2 ijerph-18-08544-t002:** Possibly problematic situations concerning drug purchasing practices.

Questions	Mean (SD)	*t* Value
P1. I often wonder which drugstore has the medicine I want to buy	2.76 (1.04)	−5.39 *
P2. It is not always known which drugstore is cheaper, more affordable, or has promotional activities	3.22 (1.06)	4.93 *
P3. When faced with the same drugs produced by different manufacturers, I often do not know which one to buy	3.36 (1.08)	7.87 *

* *p* < 0.01.

**Table 3 ijerph-18-08544-t003:** Possibly problematic situations concerning the drug use practice.

Questions	Mean (SD)	*t* Value
U1. When I decide to take drugs, I often have problems such as how to combine various drugs, their dosages, and the times I take the drugs	2.68 (1.03)	−7.87 *
U2. In the later period of medication or after getting better, it is often uncertain whether the drugs need to be adjusted, such as reducing the dose, choosing other alternative drugs, and so on	3.15 (0.97)	3.58 *
U3. In the long term, I often forget to take drugs on time	3.06 (1.04)	1.39

* *p* < 0.01.

**Table 4 ijerph-18-08544-t004:** Practices and problems/opportunities.

Practices	Questions	Problems/Opportunities
Selection	How	Most consumers depend on their experience and recommendations by sales personnel
Purchasing	Where	Consumers do not know which drugstores offer cheaper drugs or discount activities
	What	It is difficult for consumers to choose between manufacturers
Use	How to adjust	Consumers do not know how to adjust their medication
Storage	Why	Habit of storing large amounts of drugs and buying drugs in large package sizes
	How to clear up	Most families clear up drugs cabinets irregularly and have large amounts of expired drugs in their homes.
Disposal	How to recycle	Most consumers do not know about the activity while a significant share of consumers have actually engaged in recycling activities
	How to dispose	Most consumers choose to throw drugs into the garbage

## Data Availability

The data presented in this study are available on request from the corresponding author.

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
