# Peer review of "Household Drug Management Practices of Residents in a Second-Tier City in China: Opportunities for Reducing Drug Waste and Environmental Pollution"

_ijerph, 2021, doi:10.3390/ijerph18168544_

Round 1

Reviewer 1 Report

The present study considers household drug management practices as five sub-practices: selection, purchasing, use, storage and disposal and explains how it was carried out among Chinese 80 households to assess them. However, in order to make it publishable, I suggest the authors to consider the following:

Shape suggestions

Please English revising - regarding repetition (L16, L18. We...; L151, L152, L157. We.... (please avoid personal manner of addressing and use the impersonal manner (i.e.  It was conducted), the text will sound much more professional; there are so many "We" through the manuscript. L81, L84. The findings...Please reshape and revise the entire manuscript in these 2 regards.

Figure 1. Please write "Purchasing" in a single line.

Please remove the item "the" before the following: L94. "how"; L103. "which"; L112. "how"; L119. "why" and "how"; L138. "how".

Table 1. Please consider the following aspects:

  • At Age, please mention (Years) as for the Income you mentioned (RMB) - what is this?. Please explain each abbreviation in full at its first usage in the manuscript and under each figure/table (please check the Instructions for the authors https://www.mdpi.com/journal/ijerph/instructions ).
  • Frequency in what? Please complete
  • Last column: please add in the head of the table, under Percentage, the symbole % and remove it near each numerical value in that column.

Figures 4-8 are blurred. Please replace them with better ones. I suggest also to group figures 3-4, 5-7, and 8-9 in single lines, not a figure under the other (tabulate them, with no visible line). The text will look much more condensed and representative. In its current form, there is far too much empty space.

Title of Table 4. L 287. Practices not practices

Content suggestions

Please extract the aim of the study (from the last paragraph of Introduction section), making it a separate paragraph, highlighting as better is possible the novelty character of the paper or the special aspects that it brings to the field (in the topic) so as to attract the best possible attention of those interested. 

L 75-76. The authors have mentioned Some studies... However, few studies...

Indeed, there are a lot of studies considering the topic the authors cited, but they must be mentioned to show that there are many. In this regard I suggest as follows: Tit D.M. et al. Disposal of unused medicines resulting from home treatment in Romania, J. Environ. Prot. Ecol. 2016. 17(4), 2016, 1425-1433; Bungau S.et al. Studies about last stage of product lifecycle management for a pharmaceutical product, J. Environ. Prot. Ecol. 2015. 16(1), 2015, 56-62; Bungau, S. et al.  Aspects regarding the pharmaceutical waste management in Romania, Sustainability, 10(8), 2018, 2788. https://doi.org/10.3390/su10082788

Paragraph L133-138 must be referenced. In this regard, please check the following two references: https://doi.org/10.1016/j.coesh.2020.10.012 https://doi.org/10.3390/antibiotics.9020081

Section 3.1. How were the questionnaires thought? If specialists in the field of pharmacy / environment have been consulted? How were the items of the questionnaires chosen? Who validated the questionnaires? What if there was a pretest of them regarding the relevance of the questions and the potential respondents? Is the group of respondents relevant in correctly characterizing the population of Kunming city?

Author Response

Below we provide point-by-point responses. Please see the attachment

Reviewer 2 Report

Thank you for the opportunity to review your manuscript.

Well-written outline of issues arising from “Household Drug Management Practices of Residents in a Second-tier City in China: Opportunities for Reducing Drug Waste and Environmental Pollution”.

I have read through the paper with interest with some issues to be addressed by the authors:

I wanted to comment on a few perceived limitations of the study that you would hopefully consider addressing. Although it would likely require a larger sample size to have adequate statistical power, which is important considering that a randomized, controlled trial is not feasible. A second perceived limitation is the enrollment strategy. Without any form of randomization of potential study participants or blinding, this method is highly susceptible to selection bias. I would recommend that there be a more systematic and fair approach to enrollment.

Reassess the Likert scale used (5-point Likert scale). The seven-point scale usually reaches the upper limit of the scale’s reliability. Try as far as possible to use a wide scale. For your analysis, which will take place at a later stage, you can always condense the responses. This reviewer would argue in favor of the 7-point Likert scale since it is believed to give a more precise measure of the participant’s evaluation.

Author Response

Below we provide point-by-point responses. Please see the attachment.

Reviewer 3 Report

Minor editing

Don’t start sentence with a number, please write word for 17α-

Write full names of abbreviations when first mentioned

Please provide reference for the following statement

From a consumer perspective, storage concerns the ‘why’ and the ‘how’. ‘Why’ refers 119 to why consumers store medicines in their homes. In China, storing drugs for emergent 120 use in homes in large quantities and varieties has become a common practice.

Please describe the available options for drug disposal in the setting of the study

Please stress whether the manuscript/questionnaire refers to otc drugs (self-medication) or all drugs. Was this explained to the participants? If not please add to limitations

Using items of a a not-validated questionnaire for predictions such as in table 2 is greatly optimistic and should be discouraged and rewritten. Please use descriptive analysis or provide detailed explanation of statistical analysis used.

Did any of the questions allow for additional answers? I.e. Reasons to store drugs at home

If possible provide the questionnaire as supplementary data

Author Response

Thanks a lot. Below we provide point-by-point responses. Please see the attachment.

Round 2

Reviewer 1 Report

The authors responded to all my requests.